# Expression of *GLOD4* in the Testis of the Qianbei Ma Goat and Its Effect on Leydig Cells

**DOI:** 10.3390/ani14172611

**Published:** 2024-09-08

**Authors:** Jinqian Wang, Xiang Chen, Wei Sun, Wen Tang, Jiajing Chen, Yuan Zhang, Ruiyang Li, Yanfei Wang

**Affiliations:** 1Key Laboratory of Animal Genetics, Breeding and Reproduction in The Plateau Mountainous Region, Ministry of Education, Guizhou University, Guiyang 550025, China; 15114781228@163.com (J.W.); went4863@gmail.com (W.T.); chenjiajing511@163.com (J.C.); zhangyuan990428@163.com (Y.Z.); l549091621@163.com (R.L.); yfwang202206@163.com (Y.W.); 2College of Animal Science, Guizhou University, Guiyang 550025, China; 3International Joint Research Laboratory in Universities of Jiangsu Province of China for Domestic Animal Germplasm Resources and Genetic Improvement, Yangzhou University, Yangzhou 225009, China; dkxmsunwei@163.com; 4College of Animal Science and Technology, Yangzhou University, Yangzhou 225009, China

**Keywords:** Qianbei Ma goat, testosterone, proliferation, Leydig cell, *GLOD4*

## Abstract

**Simple Summary:**

Glyoxalase domain-containing protein 4 (*GLOD4*), a member of the glyoxalase gene family, is widely expressed in Leydig cells and its primary function within the Leydig cells remains unknown. *GLOD4* expression in other species is often associated with functions such as cell proliferation, cell cycle, and detoxification. We therefore explored the relevant elements of Leydig cells. We speculate that *GLOD4* may regulate testicular development and spermatogenesis by increasing the number of Leydig cells and regulation of testosterone secretion.

**Abstract:**

The expression pattern of *GLOD4* in the testis and its regulatory effect on testicular cells was explored in goats to enhance our understanding of spermatogenesis and improve reproduction in breeding rams. In this study, we demonstrated the localization of *GLOD4* in testicular cells using immunohistochemistry and subcellular localization analyses. Subsequently, we analyzed the *GLOD4* expression pattern in four age-based groups (0, 6, 12, and 18 months old) using real-time quantitative polymerase chain reaction (qRT-PCR) and protein blotting. Finally, we performed *GLOD4* silencing and overexpression studies in Leydig cells (LCs) and explored the effects on cell proliferation, the cell cycle, steroid hormone secretion and the expression of candidate testosterone hormone-regulated genes. *GLOD4* was mainly expressed in Leydig cells, and the subcellular localization results showed that the GLOD4 protein was mainly localized in the cytoplasm and nucleus. Silencing of *GLOD4* significantly suppressed the mRNA expression levels of the testosterone secretion-related genes *CYP11A1*, *3β-HSD*, and *CYP17A1* and the mRNA expression levels of cell cycle-related genes *CDK6*, *PCNA*, and *Cyclin E*. Moreover, the cell cycle was blocked at the G2/M phase after *GLOD4* silencing, which significantly suppressed testosterone secretion. In contrast, *GLOD4* overexpression significantly increased the mRNA expression levels of the testosterone secretion-related genes *CYP11A1*, *3β-HSD*, and *CYP17A1* and increased the expression of the cell cycle-related genes *CDK6*, *PCNA*, and *Cyclin E*. Moreover, *GLOD4* overexpression promoted the cell cycle from G0/G1 phases to enter the S phase and G2/M phases, promoted the secretion of testosterone. Taken together, our experimental results indicate that *GLOD4* may affect the development of cells in Qianbei Ma goats of different ages by influencing the cell cycle, cell proliferation, and testosterone hormone synthesis. These findings enhance our understanding of the functions of *GLOD4* in goats.

## 1. Introduction

In the testis, spermatogenesis begins with the mitotic division of spermatogonia in the seminiferous tubules. Spermatogonia undergoes two meiotic divisions to form spermatocytes, which subsequently undergo a complex series of changes to produce mature spermatozoa. This highly organized process is regulated by many highly complex genes [1,2,3,4,5]. Spermatogenesis is a highly complex and organized process that occurs in the testis. Another important biological function of the testis, which is an important male reproductive organ, is androgen secretion. For example, testosterone increases the libido and sexual behavior, improves spermatogenesis and sperm quality in male animals, increases the success rate of fertilization [6], and can induce the testicular tubules to produce more sperm and support sperm maturation and motility [4,7]. In addition, estrogen and other hormones such as luteinizing hormone (LH) and follicle-stimulating hormone (FSH) are crucial throughout spermatogenesis [8,9]. Leydig cells are in the interstitial tissue of the testes and play an important role in the male reproductive system. One of the main functions of Leydig cells is to secrete testosterone, a hormone that stimulates sperm production and development within the testes [10,11]. Since the peak of testosterone secretion mainly occurs during the reproductive and youthful stages of the animal, testosterone secretion also has an indispensable role in muscle protein synthesis, calcium uptake and bone density maintenance during these developmental stages. In addition to testosterone secretion, Leydig cells also regulate the proper testis temperature and the proliferation and function of different cell types within the testis, thus ensuring normal spermatogenesis [12]. As a domesticated animal with important economic value, the economic impact of goats can be increased by improving the fertility of breeding rams and the efficiency of goat breeding [13]. Recent studies have revealed that GLOD4 is differentially expressed in the Leydig cells of Tibetan sheep at different developmental stages [14].

The glyoxalase gene family is an important regulatory family consisting of six structurally and functionally distinct enzymes, including glyoxalase domain-containing protein 4 (*GLOD4*). *GLOD4*, also known as HC6-type or HC71-type, was originally cloned as the *C17orf25* cDNA from human hepatocellular carcinoma [15,16]. The human *GLOD4* region is approximately 23 kb long and contains 10 exons and 9 introns. The full-length cDNA sequence of human *GLOD4* is 1814 bp long and encodes an open reading frame *(ORF*) consisting of 313 amino acids [17]. The downregulation of *GLOD4* expression in the mouse embryonic stem cell is thought to inhibit cell proliferation and block the cell cycle [18]. In a canine model of retinal degeneration caused by mutations in the *RPGRORF15* gene, researchers analyzed the expression profiles of mutant and normal retinas. In conjunction with other genes, *GLOD4* was found to have an effect on cell cycle progression and to potentially influence processes related to mitochondrial function and cellular energy metabolism [19]. In Blmh (Bleomycin Hydrolase)−/−mice, *GLOD4* isoform 3 mRNA expression was decreased in the kidney but no effect on isoform 1 mRNA expression was observed. This suggests that GLOD4 protein expression may be subjected to Blmh+/+-mediated posttranscriptional regulation, which, in turn, has a detoxification effect [20]. Using the nematode Caenorhabditis elegans (*C. elegans*) as a model to explore the role of the TRPA1-Nrf signaling pathway in methylglyoxal detoxification revealed that it regulates the expression of *GLOD4*. This study also emphasized the potential role of *GLOD4* in human diseases, especially in pathologies associated with α-diketone bodies’ (α-DCs) accumulation, such as diabetes and neurodegenerative diseases [21]. However, current *GLOD4* research still focuses mainly on humans and rodents, with few reports on goats.

However, the mechanism by which *GLOD4* expression affects Leydig cells in goats has rarely been reported, especially within the context of testicular development. As a goat breed that is well adapted to the highland mountain environment, the Qianbei Ma goat has a long history of breeding in Guizhou Province with a medium body size, rough feeding resistance, strong disease resistance, good fur quality, and strong fertility. Thus, the Qianbei Ma goat is a good model animal for exploring the reproductive traits of plateau goats [22]. Therefore, the aim of this study was to investigate the profile of *GLOD4* expression in the testes of goats at different ages and to analyze the regulatory effect of *GLOD4* expression in Leydig cells. We silenced and overexpressed *GLOD4* in the Leydig cells of goats and investigated *GLOD4* expression in Leydig cells. This study established a foundation for understanding the potential regulatory mechanism through which the *GLOD4* functions in spermatogenesis and provided a theoretical basis for improving the reproductive traits of breeding rams.

## 2. Materials and Methods

The experimental protocol used in this study was approved by the Animal Protection and Use Committee of Guizhou University, Guiyang, China (Approval number: EAE-GZU-2021-E021). The animal handling procedures were in line with the Chinese Animal Welfare Guidelines and were approved by the Animal Protection and Use Committee of Guizhou University, Guiyang, China (Approval number: EAE-GZU-2021-E021).

### 2.1. Tissue Collection

A total of 20 healthy goats were randomly selected from Xishui County, Guizhou Province, China, Fuxing Dairy Company Limited, and divided into four age groups (*n* = 5 for each age group): 0 M, 0 month old (before sexual maturity, body weight: 1.66 kg ± 0.5 kg); 6 M, 6 months (after sexual maturity, body weight: 30.30 kg ± 1.0 kg); 12 M, 12 months old (after physical maturity, body weight: 43.32 kg ± 1.5 kg); 18 M, 18 months (sell on the market, body weight: 47.53 kg ± 2.0 kg). After anesthesia, the left testicular tissue was castrated and immediately collected with sterile scissors forceps and trimmed to approximately 5 mm testicular tissue fixed in 4% paraformaldehyde solution for H&E staining, with three biological replicates at each age. Testicular tissue samples were collected and washed with phosphate-buffered salt solution within 20 min. Then, all samples were snap frozen in liquid nitrogen and subsequently transferred to −80 °C for storage for further studies.

### 2.2. Cell Culture and Transfection

Fine Leydig cells were obtained from 12-month-old goat testes and cultured as previously described [23]. Briefly, the testicular tissue from goats was immediately transferred to Dulbecco’s phosphate-buffered saline (DPBS) supplemented with 5% penicillin/streptomycin (Gibco, Grand Island, NY, USA) and incubated at 4 °C. The white membrane on the surface of the testis was removed with eye clippers and then digested with 0.5 mg/mL collagenase IV (Sigma, St. Louis, MO, USA) in DMEM/F12 (Gibco, Grand Island, NY, USA) at 34 °C for 20 min. The dispersed cells were then filtered using a 100 mm cell strainer (Biosharp, Shanghai, China), and the filtrate was centrifuged at 1200× *g* for 5 min. The supernatant was discarded, and the cells were washed three times in DPBS and centrifuged at 1200× *g* for 5 min and resuspended. The resuspended cells were purified by the differential applanation method. The harvested Leydig cells were cultured in DMEM/F12 containing 10% fetal bovine serum (Gibco, Beijing, China) and 1% penicillin/streptomycin and incubated in a cell culture incubator (Thermo Fisher Scientific, Waltham, MA, USA) at 34 °C with 5% CO_2_. Once the confluence reached 75–90%, the cells were evenly divided between 6 empty plates (NEST Biotechnology Co., Ltd., Wuxi, China). These cells were then transfected with *GLOD4* overexpression and silencing plasmids using a Lipofectamine 2000 kit (Thermo Fisher Scientific Waltham, MA, USA). After 48 h, the RNA luminescence was assessed as good, and the cells were harvested to conduct all other experiments.

### 2.3. Plasmid Construction

The PCR reaction (25 µL) consisted of 1 µL of DNA template, 1 µL of each of the upstream and downstream primers (10 pmol/µL), 12.5 µL of Green Mix, and 9.5 µL of ddH2O. After PCR amplification, the amplicons were subjected to gel purification to obtain the desired target sequences, which were then ligated to pcDNA3.1 using T4 DNA ligase. Plasmids were amplified in transformed DH5αcompetent cells. Then, the desired plasmids were extracted from the cells by oscillation. sh−*GLOD4*-interfering sequences were designed and synthesized by Shanghai Gemma Pharmaceuticals & Biologicals (see Table 1 for details).

### 2.4. Total RNA Was Extracted and Reverse-Transcribed

The total RNA was extracted from the goat testicular tissue and Leydig cells using TRIzol reagent (Solarbio, Beijing, China). The total RNA extracted for each project was more than 30ug. Subsequently, first-strand cDNA was performed using the StarScript II First-Strand cDNA Kit (GenStar, Beijing, China) according to the manufacturer’s instructions, and the reaction products were stored at −20 °C.

### 2.5. Real-Time Polymerase Chain Reaction

The expression levels of *GLOD4*, and that of other genes, were measured in the goat Leydig cells on a CFX 9600 real-time PCR instrument (Bio-Rad, Hercules, CA, USA). The details of the real-time PCR primers used are summarized in Table 2. A 10 µL real-time PCR mixture (containing 5 µL of 2× Es-Taq Master mix, 3.2 µL of RNase-free ddH2O, 0.5 µL of cDNA, and 0.4 µL of forward and reverse primers (10 pmol/µL)) was prepared according to the manufacturer’s instructions and subjected to a three-step PCR program (95 °C for 2 min, 95 °C for 15 s, annealing temperature (see Table 2 for details) for 30 s, and 72 °C for 30 s; the melting curve was generated automatically by the machine). The PCR assay was repeated three times for each test sample. The expression data were normalized to the expression of β-actin, which we used as an internal reference. The relative change in gene expression was then calculated using the relative quantification method 2^−∆∆CT^. β-actin primers were synthesized by Bioengineering (Shanghai) Co. (see Table 2 for details).

### 2.6. Western-Blot

Testicular tissues from different developmental stages were homogenized and lysed using a radioimmunoprecipitation assay (RIPA) protein extraction kit (Solarbio, Beijing, China) with the addition of phenylmethylsulfonyl fluoride (PMSF) (Solarbio, Beijing, China). Protein concentrations were quantified using a commercial bicinchoninic acid (BCA) protein assay (Solarbio, Beijing, China). Protein samples were separated on a 10% SDS-PAGE gel and then transferred to a polyvinylidene fluoride (PVDF) membrane by wet transfer. After being blocked in 5% skim milk for 2 h, the membranes were incubated overnight with rabbit anti-*GLOD4* polyclonal antibody (1:1000 dilution, Bioss, Beijing, China) at 4 °C. The membranes were then washed three times in Tris-buffered saline containing Tween-20 (TBST), incubated with an HRP-conjugated secondary antibody (1:10,000 dilution) for 1 h in TBST, washed again, and then incubated with an HRP-conjugated secondary antibody (1:10,000 dilution, Bioss, Beijing, China) and visualized using an NcmECL Ultra kit (P10200, NCM, Suzhou, China) on a Universal Hood II instrument (Bio-Rad, Hercules, CA, USA). The grayscale value of each band was analyzed using ImageJ software (National Institutes ofHealth, Bethesda, MD, USA).

### 2.7. Immunohistochemistry

For immunohistochemistry, after the prepared paraffin sections of testis were dewaxed in water, they were boiled in 0.1 M citrate buffer (pH = 6.0) for 15 min in a microwave oven and then cooled to 4 °C for antigen retrieval. The sections were subsequently washed with phosphate-buffered saline (PBS) and incubated in 3% hydrogen peroxide for 15 min to block endogenous peroxidase activity. The sections were further incubated in blocking buffer (5% bovine serum albumin (BSA) in PBS) for 15 min at 34 °C to block nonspecific binding. The membranes were then incubated with rabbit anti-*GLOD4* antibody (1:1000; Bioss Biotechnology Co., Ltd., Beijing, China) in PBS overnight, washed with PBS, incubated with the biotinylated secondary antibody for 2 h at 34 °C and with horseradish peroxidase (HRP)-streptavidin for 15 min, and then visualized using diaminobenzidine. Nonspecific rabbit immunoglobulin G (IgG) was used instead of the primary antibody as a negative control.

### 2.8. Subcellular Localization

After transfecting the overexpressed plasmids containing GFP fluorescence for 36 h, the cells were washed 3 times with PBS and fixed at room temperature for 20 min by the addition of precooled 4% paraformaldehyde (Solarbio, Shanghai, China); the cells were washed 3 times with PBS and fixed at room temperature for 5 min by the addition of 0.25% Triton X-100 (Biosharp Anhui, China); to conduct all other experiments, and the nuclei were stained with DAPI (Solarbio, Shanghai, China) to stain the cells; and the cells were washed 3 times with PBS. The fluorescence images were merged using Photoshop, and the subcellular localization of the fusion protein was assessed based on the overlap between the fluorescence of the fusion protein and the fluorescence of the cell nucleus.

### 2.9. Cell Proliferation Analysis

The CCK-8 assay was used to detect the effect of *GLOD4* overexpression and silencing on the proliferative activity of Leydig cells. Leydig cells grown in the logarithmic growth phase were selected, digested with trypsin, and counted. Cell suspensions were added in 96-well plates at 100 μL/well. Plasmids from the negative control and experimental groups were incubated at 34 °C and 5% CO_2_ for 6, 12, 24, 48 and 72 h after transfection, as described in the instructions of the CCK-8 assay kit (Monmouth Junction, NJ, USA). Five replicates were used within each experimental and control group for each of the five time periods. The absorbance (OD) at 450 nm was subsequently measured using enzyme markers (Thermo Fisher Scientific, Waltham, MA, USA).

### 2.10. Steroid Assay

The culture medium was collected to measure the testosterone concentration using an ELISA kit (Kamels, Shanghai, China) according to the manufacturer’s instructions [24]; three replicates within each experimental and control group for each of the two time periods. The testosterone concentration was examined at 48 h and 72 h after transfection of the interference and overexpression plasmids. The absorbance of the samples was measured at 450 nm, and the concentration of testosterone was determined using a standard curve.

### 2.11. Flow Cytometry Analysis

The harvested cells were fixed in 70% ice-cold ethanol overnight at 20 °C. After washing with PBS, the cells were incubated in 0.5 mg/mL RNase for 30 min at 35 °C and stained with 0.025 mg/mL propidium iodide for 10 min. Finally, cells at different stages of the cell cycle were identified by flow cytometry analysis (BD Biosciences, Canto II plus, NY, USA). The data from at least 20,000 cells were collected per sample.

### 2.12. Statistics

Statistical analyses were performed using Graphpad prism10. All measurements were repeated at least three times, and the data were presented as means ± SD. The mRNA and protein expression of GLOD4 in the testicular tissues of a Qianbei Ma goat at different months of age was analyzed using one-way ANOVA, two-way ANOVA was used for the CCK-8 experiment, and *t*-test was used for all the rest of the data results. (significant, * *p* < 0.05; highly significant, ** *p* < 0.01).

## 3. Results

### 3.1. Localization of GLOD4 in the Testis of Qianbei Ma Goat

We detected GLOD4-positive protein signals by immunohistochemistry in testicular tissue sections from 12-month-old rams and found that the immunopositivity was confined to Leydig cells (Figure 1A). After transfection of the pcDNA3.1−NC plasmids containing GFP fluorescence, the subcellular localization results showed that *GLOD4* was highly expressed in the nucleus and cytoplasm of Leydig cells (Figure 1B).

### 3.2. Expression Patterns of the GLOD4 at Different Developmental Stages in Qianbei Ma Goat Testicles

Both the Western blot and qRT-PCR showed that *GLOD4* was expressed in the testes of Qianbei Ma goats at 0, 6, 12 and 18 months of age, and *GLOD4* expression increased with age. GLOD4 protein expression was significantly higher at 6 and 12 months of age than at 0 months of age (*p* < 0.05). The *GLOD4* mRNA expression was significantly higher at 6, 12 and 18 months of age than at 0 months of age (*p* < 0.01; Figure 2A,B).

### 3.3. Effect of GLOD4 on the Proliferation of Goat Leydig Cells

After *GLOD4* silencing and overexpression, both the mRNA and protein levels of *GLOD4* were significantly downregulated and upregulated (*p* < 0.01 each), respectively, compared to those in the control group (sh−NC, pcDNA3.1−NC) (Figure 3A–D). The CCK-8 assay at 6, 12, 24, 48 and 72 h after transfection, which revealed that silencing of *GLOD4* inhibited the proliferation of Leydig cells, whereas *GLOD4* overexpression significantly promotes cell proliferation. (Figure 3B,C). In addition, we found that *GLOD4* interference and overexpression resulted in the downregulation and upregulation of *PCNA*, respectively (Figure 4A).

### 3.4. Effect of GLOD4 on Cell Cycle Progress of Goat Leydig Cells

The percentage of cells in the G2/M phase was (*p* < 0.05) lower than that in the control group after *GLOD4* silencing and expression of the cell cycle-regulated protein kinase (*CDK6*) and *Cyclin E* (*Cyclin E*) was significantly reduced in these cells (*p* < 0.05) (Figure 5A,B). In contrast, overexpression of *GLOD4* significantly reduced the percentage of cells arrested in the G0/G1 phase and increased the percentage of cells in the G2/M phase compared to those in the control group, and highly significantly (*p* < 0.01) upregulated the expression levels of *CDK6* and *Cyclin E* (Figure 5C,D).

### 3.5. Promotion of Testosterone Hormone Secretion by GLOD4

After *GLOD4* silencing, it secreted less testosterone (*p* < 0.01) than that in the controls at 48 and 72 h, and the expression of *3β-HSD* (*p* < 0.05) and *CYP17A1* (*p* < 0.01) significantly decreased (Figure 6A,C). In addition, after overexpression of *GLOD4*, Leydig cells secreted (*p* < 0.01) more testosterone hormone than the control group at 48 h and 72 h, significantly increased expression of *3β-HSD* (*p* < 0.01) and *CYP17A1* (*p* < 0.05), and had no effect on *CYP11A1* expression (Figure 6B,D).

## 4. Discussion

Previous reports on *GLOD4* have focused on its metabolic function and detoxification properties [21,25,26], although some have reported that *GLOD4* may have certain effects on the cell cycle and other cellular functions [18,27]. However, few studies have explored the role of *GLOD4* in reproduction. Therefore, by exploring *GLOD4* expression in the testis of the Qianbei Ma goat and its effect on Leydig cells, we provided insights about the function of *GLOD4* and a theoretical basis for improving ram fertility.

Wang et al. found that *GLOD4* expression was significantly upregulated with age in the testis of Tibetan sheep, and *GLOD4* was only expressed in Leydig cells and that *GLOD4* was mainly observed in the cytoplasm [22]. Our immunohistochemistry revealed that *GLOD4* was expressed only in Leydig cells; however, ours revealed expression in both the nucleus and the cytoplasm. Interestingly, in a recent study, fluorescently tagged GLOD4 protein was widely distributed in the cytoplasm and nucleus of human retinal pigment epithelial cells (*hTERT-RPE1*) [27], which may result from differential gene expression in different species [28,29,30].

Based on the differential expression of *GLOD4* at different testicular development stages, we hypothesized that the observed increase in *GLOD4* expression might relate to the continuous development of the testis. Previously, using a yeast two-hybrid method, Zhang et al. screened for *NUDT9* interactors and reported that *GLOD4* may inhibit cell growth by interacting with *NUDT9* [31]. In a recent report on *GLOD4*, *GLOD4*-deficient Hidradenitis elegans nematode worms were shown to exhibit increased α-DCs, a shortened lifespan, increased neuronal damage, and tactile sensitization, indicating that *GLOD4* expression has a protective effect on the cells and ensures cellular proliferation [26]. We found that *GLOD4* silencing significantly inhibited cell proliferation at two time points and downregulated the mRNA expression of the cellular nuclear antigen (*PCNA*), which is a proliferation marker. *PCNA* is widely distributed in a variety of cells, reflecting not only their proliferation rate but also affecting cellular proliferation as a key gene [32,33]. In contrast, overexpression of *GLOD4* had a significant proliferation effect and significantly upregulated the expression of *PCNA*. Therefore, we speculate that *GLOD4* may play an important regulatory role in cell proliferation.

The cell cycle, which is regulated by the cell cycle proteins CDK and CKI, controls cell growth and plays a key role in cell proliferation [34,35,36]. Our results showed that both *GLOD4* silencing and overexpression significantly inhibited and promoted the progression of the cell cycle from the S phase to the G2/M phase, respectively. Similarly, Dihazi et al. asserted that downregulation of GLOD4 protein expression may inhibit cellular proliferation and block the cell cycle in mouse embryonic stem cells [18]. In a genome-wide study of *Chlamydomonas reinhardtii* cilia, Albee Alison et al. identified *GLOD4* as one of the genes whose expression is upregulated during ciliogenesis. Moreover, knockdown of *GLOD4* in human retinal pigment epithelial cells (hTERT-RPE1) results in shorter cellular cilia and cell cycle progression defects [27]. In our experiment, silencing and overexpression of GLOD4 significantly downregulated and upregulated both CDK6 and Cyclin E, respectively. CDK6 is a key gene in the CDK family which regulates cell cycle progression from the G1 to the S phase by phosphorylating retinoblastoma (RB) proteins when it is complexed with cyclin D [37]. In addition, Cyclin E performs a similar function via regulating the cellular transition from the G0/G1 phase to the S phase to promote cell cycle progression [38]. In addition, overexpression of GLOD4 accelerated the progression from G0/G1, and we speculated that CDK6 and Cyclin E might play a role in this. Therefore, we speculate that GLOD4 may regulate cell proliferation by affecting the G2/M phase of the cell cycle.

Taken together, *GLOD4* may be crucial for Leydig cells’ proliferation, but how *GLOD4* functions in the regulation of testosterone secretion remains unknown. To address this, we examined testosterone secretion in cells cultured for different durations and the expression of genes involved in testosterone secretion. It is well known that in mammalian testes, the testicular cellular receptor (*SCARB1*) binds specifically to the raw material for androgen synthesis [39], and a series of cascade reactions stimulate the expression and phosphorylation of steroidogenic acute regulatory protein (*STAR*) [40]. In turn, *STAR* catalyzes the substrate of the cytochrome P450 cholesterol side chain cleavage enzyme (*CYP11A1*) reaction and converts it to pregnenolone, which is then catalyzed to progesterone by the micronutrient enzyme 3B-hydroxysteroid dehydrogenase 1 (*3β-HSD*1) in the smooth endoplasmic reticulum [41]. Progesterone is ultimately converted to testosterone by a series of steroid synthases [42,43,44]. These functional genes play a critical role in regulating testosterone secretion. *3β-HSD* and *CYP17A1* were significantly downregulated and upregulated, respectively, by *GLOD4* suppression and overexpression. In addition, the suppression and overexpression of *GLOD4* decreased and increased testosterone secretion in Leydig cells at 48 and 72 h, respectively. These results suggest that *GLOD4* influences testosterone secretion by affecting the expression of a range of genes involved in steroid synthesis.

## 5. Conclusions

In summary, *GLOD4* was differentially expressed in the testes of goats at different ages. In addition, after the silencing and overexpression of *GLOD4* in goats, the proliferation of Leydig cells was decreased and increased, respectively; the transition from the S phase to the G2/M phase was blocked and accelerated; and testosterone secretion was significantly decreased and increased, respectively. Therefore, our results suggested that *GLOD4* may indirectly regulate testicular development and spermatogenesis by increasing the number of Leydig cells and regulation testosterone secretion, indicating that *GLOD4* plays a key role in spermatogenesis by affecting the fate of Leydig cells in goats. Although our study confirms the value of exploring the role of *GLOD4* in reproduction, there remains a large gap in studies exploring the role of this gene in reproduction, and larger studies with longer follow-up periods are needed to explore the mechanism by which *GLOD4* regulates testicular development in vitro and in vivo.

## Figures and Tables

**Figure 1 animals-14-02611-f001:**
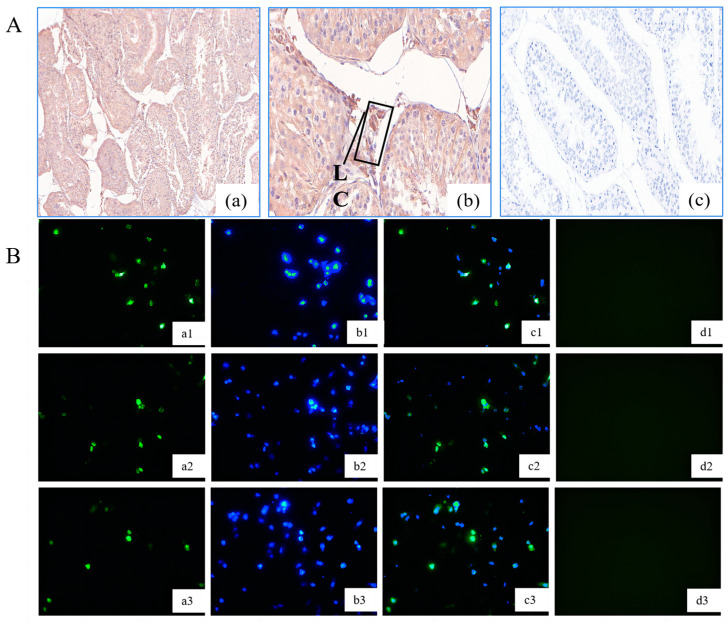
*Localization of* GLOD4 protein in 12-month-old testis tissue and Leydig cells. (**A**) Immunohistochemical staining of GLOD4 protein in the testes of a 12-month Qianbei Ma goat. (**a**) Immunostaining pattern of GLOD4 protein in the 12M testes (100×) of a Qianbei Ma goat. (**b**) Immunostaining pattern of GLOD4 protein in the 12M testes (400×) of a Qianbei Ma goat. LC, Leydig cell; (**c**) phosphate-buffered saline (PBS) was used instead of the primary antibody as a negative control (400×). (**B**) Subcellular localization of *GLOD4* on the goat Leydig cells. *GLOD4* was labeled with GFP fluorescence (**a1**–**a3**) and the nuclei were labeled with DAPI dye (**b1**–**b3**) (100×). c (**c1**–**c3**) Diagram combining a (**a1**–**a3**) + b (**b1**–**b3**); (**d1**–**d3**) blank control (*n* = 3).

**Figure 2 animals-14-02611-f002:**
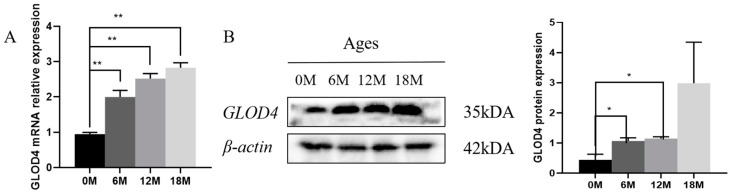
Expression of GLOD4 in testes at different months of age. (**A**) *GLOD4* mRNA expression in testes of goats of different ages. (**B**) GLOD4 protein expression in testes of goats of different ages. Data are expressed as mean ± standard deviation, * *p* < 0.05; ** *p* < 0.01.

**Figure 3 animals-14-02611-f003:**
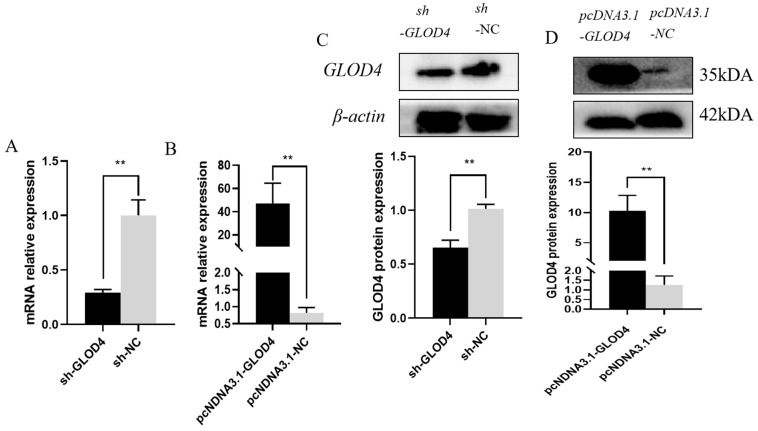
Validation of *GLOD4* overexpression and silencing expression vector efficiency. (**A**,**B**) *GLOD4* mRNA and protein expression levels in goat Leydig cells transfected with shRNA−*GLOD4* and sh−NC. (**C**,**D**) GLOD4 mRNA and protein expression levels in goat Leydig cells transfected with pcDNA3.1−*GLOD4* and pcDNA3.1−NC. Data are expressed as mean ± standard deviation, ** *p* < 0.01.

**Figure 4 animals-14-02611-f004:**
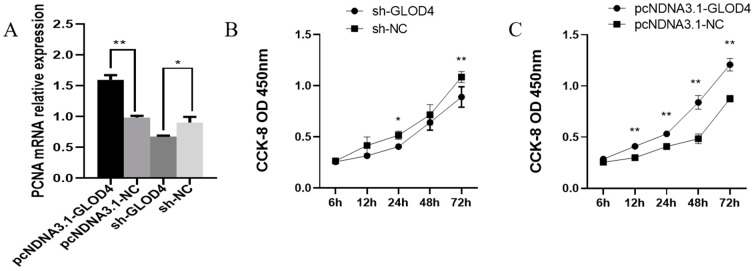
Effects of altering *GLOD4* on proliferation of Leydig cells. (**A**) mRNA expression levels of *PCNA* in goat transfected with sh−*GLOD4*, sh−NC, pcDNA3.1−NC, and pcDNA3.1−*GLOD4*. (**B**,**C**) Cell proliferation was detected by CCK−8 after silencing and overexpressing *GLOD4*. Data are expressed as mean ± standard deviation, * *p* < 0.05, ** *p* < 0.01.

**Figure 5 animals-14-02611-f005:**
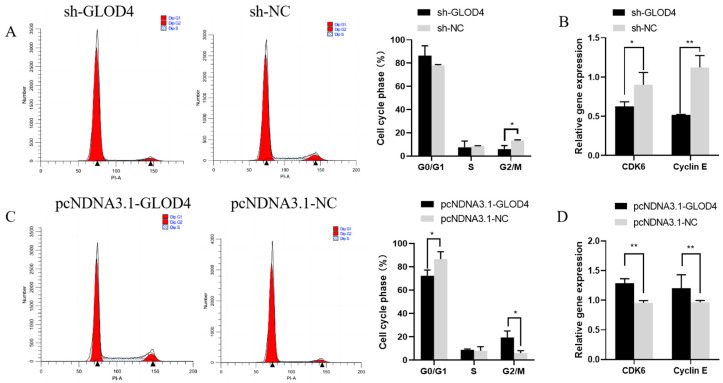
Effect of *GLOD4* on cell cycle progress of Qianbei Ma goat Leydig cells in vitro. (**A**,**C**) Effects of goat cells transfected with sh−*GLOD4* and pcDNA3.1−*GLOD4* on cell cycle progression. (**B**,**D**) Expression levels of cell cycle-related genes in Qianbei Ma goat after silencing and overexpressing *GLOD4*. All tests were performed at least three times. Data are expressed as mean ± standard deviation, * *p* < 0.05; ** *p* < 0.01.

**Figure 6 animals-14-02611-f006:**
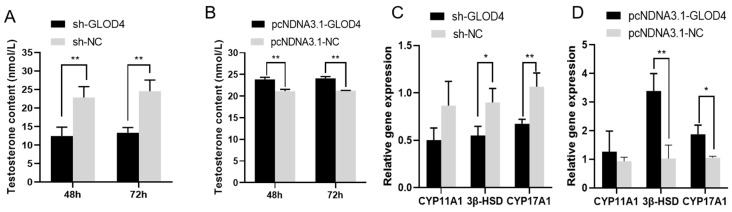
Effect of *GLOD4* testosterone hormone of Qianbei Ma goat Leydig cells in vitro. (**A**,**B**) Secretion of testosterone hormone in Leydig cells of Qianbei Ma goat transfected with sh−*GLOD4* or pcDNA3.1−*GLOD4*. (**C**,**D**) Expression levels of testosterone secretion-related genes in Leydig cells of Qianbei Ma goat after silencing and overexpressing *GLOD4*. All experiments were performed at least three times. Data are expressed as mean ± standard deviation, * *p* < 0.05; ** *p* < 0.01.

**Table 1 animals-14-02611-t001:** Plasmids construction.

Gene	Primer Sequence (5′→3′)
Sh−*GLOD4*	F:CACCGATATAAGTTCTATTTGCAGGATTCAAGAGATCCTGCAAATAGAACTTATATTTTTTTG R:GATCCAAAAAAATATAAGTTCTATTTGCAGGATCTCTTGAATCCTGCAAATAGAACTTATATC
Sh−NC	F:CACCGTTCTCCGAACGTGTCACGTTTCAAGAGAACGTGACACGTTCGGAGAATTTTTTG R:GATCCAAAAAATTCTCCGAACGTGTCACGTTCTCTTGAAACGTGACACGTTCGGAGAAC

**Table 2 animals-14-02611-t002:** Real-time fluorescent primers information.

Gene	Primer Sequence (5′→3′)	Gen Bank ID	Fragment Size (bP)	Tm/°C
*CYP17A1*	F:GCTCACCCTCGCCTATTTATT R:GTCTCCTGACACTGCTCACA	NM_001314145.1	169	58
*CYP11A1*	F:CTCCAGAGGCAATAAAGAA R:TCAAAGGCAAAGTGAAACA	NM_001287574.1	145	60
*3β* *-* *HSD*	F:AGACCAGAAGTTCGGGAGGAA R:TCTCCCTGTAGGAGTTGGGC	NM_001285716.1	292	60
*GLOD4*	F:AGCTCTGCACTTCGTGTTCA R: GCAATGCGTCCAAAACCTGT	XM_013971906.2	86	60
*CDK6*	F:GTGGACCTCTGGAGCGTTGG R:TGCCTTGCTCATCAATGTCTGTTAC	XM_018047426.1	223	58
*PCNA*	F:GTAGCCGTGTCATTGCGACTCC R:GCTCTGTAGGTTCACGCCACTTG	XM_005688167.3	145	60
*Cy* *c* *linE*	F:GATGTCGGCTGCTTAGAAT R:GTCTCCTGACACTGCTCACA	XM_018062248.1	104	60
*β-actin*	F:TGATATTGCTGCGCTCGTGGT R:GTCAGGATGCCTCTCTTGCTC	XM_018039831.1	189	60

## Data Availability

The original contributions presented in the study are included in the article and Appendix A, further inquiries can be directed to the corresponding author.

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
