# Peer review of "Expression of GLOD4 in the Testis of the Qianbei Ma Goat and Its Effect on Leydig Cells"

_animals, 2024, doi:10.3390/ani14172611_

Round 1

Reviewer 1 Report (New Reviewer)

Comments and Suggestions for Authors

Thank you for the opportunity to review this paper. I enjoyed reading about your research which has provides valuable insights into GLOD4 in the testicles of goats. 

I am unsure as to why I could see track changes in the document I reviewed and wonder if the correct version was submitted.

General comments.

The English needs refining in a number of places. Please see the PDF for the suggested emendations.

In italics GLOD4  is the abbreviation for the gene and should not have the word gene after it. If not in italics, GLOD4 is the abbreviation for the protein. Corrections in line with this should be made throughout the paper.

The abbreviation LCs is also used sporadically and should be removed and replaced with the words Leydig cells.

In the simple summary it is not clear what cellular value-addition is, therefore not appropriate language.

In the introduction the link between the testes and GLOD4 should be made in the first paragraph as indicated in the PDF.

More clarity is required in several places in the methods as indicated in the PDF including some explanation as to what each gene is and why it was selected for the real-time PCR experiment.

The section in the discussion about CDK6 and Cyclin E need re-writing. I have suggested some wording in the PDF

All figure captions should have abbreviation defined so that the figures can be understood independently of the text

There are inconsistencies in how the reference list is formatted. 

Other than the English language editing have no other concerns regarding the manuscript. The experimental design, data  collection, analysis and interpretation of the results all appear appropriate. 

Comments on the Quality of English Language

The English needs refining in a number of places. Please see the PDF for the suggested emendations.

Author Response

Thank you for the opportunity to review this paper. I enjoyed reading about your research which has provides valuable insights into GLOD4 in the testicles of goats. 

I am unsure as to why I could see track changes in the document I reviewed and wonder if the correct version was submitted.

Dear Editor, thank you for recognising and commenting in detail on our article. After carefully studying your comments (PDF), we have revised the article point by point, taking into account the content of the PDF. In this way, we hope to improve the English writing of our article as well as the overall logic of the article.

General comments.

The English needs refining in a number of places. Please see the PDF for the suggested emendations.

In italics GLOD4  is the abbreviation for the gene and should not have the word gene after it. If not in italics, GLOD4 is the abbreviation for the protein. Corrections in line with this should be made throughout the paper.

 Dear Editor, thank you for pointing this out. We have standardised the article's gene italics problem by removing redundant and incorrect expressions and updating the correct ones

The abbreviation LCs is also used sporadically and should be removed and replaced with the words Leydig cells.

Dear Editor, thank you for pointing this out. We refer to Leydig cells (LCs) in the abstract and replace Leydig cells with LCs in all that follows.

In the simple summary it is not clear what cellular value-addition is, therefore not appropriate language.

Dear Editor, thank you for pointing this out. In the new submission, we have replaced cellular value-addition with cell proliferation, which we hope will further clarify our meaningin line19.

In the introduction the link between the testes and GLOD4 should be made in the first paragraph as indicated in the PDF.

Dear Editor, thank you for pointing this out. In the introduction, we have merged the statements about the content of Leydig cells into the first paragraph and cited the significance of GLOD4 research in goats at the end of the first paragraph based on the PDF.

More clarity is required in several places in the methods as indicated in the PDF including some explanation as to what each gene is and why it was selected for the real-time PCR experiment.

Dear Editor, thank you for pointing this out. In the article, we mentioned the function of the genes related to cell proliferation and testosterone hormone secretion that were subjected to real-time PCR. what's more,we have also modified the content according to the PDF in line, 375、390、408。

The section in the discussion about CDK6 and Cyclin E need re-writing. I have suggested some wording in the PDF

Dear Editor, thank you for rewriting this section, we have made full reference to your content in line 390.

All figure captions should have abbreviation defined so that the figures can be understood independently of the text

Dear Editor, thank you for pointing this out. We have summarised each individual fig section in abbreviated form in the new submission, in the first sentence of each figure note.

There are inconsistencies in how the reference list is formatted. 

Dear Editor, thank you for pointing this out. In the new submission, we have also standardised the references e.g. abbreviated some journal names.

Reviewer 2 Report (New Reviewer)

Comments and Suggestions for Authors

This manuscript presents findings on the expression patterns and function of GLOD4 gene in the testis of Qianbei Ma goats. The experiment results indicate that GLOD4 may regulate testicular development and spermatogenesis by increasing the number of Leydig cells and regulation of testosterone secretion. This study provides a new insight into the regulatory mechanisms of testicular development. However, several points require attention:

1. In the introduction section, it is better to put the content of Leydig cells in front of the content of CLOD4 gene.

2. Results 1 showed the subcellular localization of GLOD4 protein in 12-month-old goats. What was the location information of GLOD4 protein in other periods?

3. In figure 1, the images labeled a2-b2-c2 and a3-b3-c3 do not appear to be from the same field of vision. The DAPI exposure level in b1 and c1 are inconsistent. Please review and adjust for consistency.

3. The text font in the figure should be consistent, such as, figure 2 and figure3.

4. The images in Figure 5A and 5C are of low resolution. Please replace the picture with higher definition.

5. Why was the function of GLOD4 gene only verified in 12-month-old Leydig cells in vitro? Clarifying the rationale for selecting this specific age.

Author Response

Please see details in the attachment.

This manuscript is a resubmission of an earlier submission. The following is a list of the peer review reports and author responses from that submission.

Round 1

Reviewer 1 Report

Comments and Suggestions for Authors

1. Please perform new experiments on primary culture of Leydig cells isolated from animals of different reproductive stages including sexually immature and sexually mature. Then the steroidogenic capacity and activity of Leydig cells at different reproductive stages were observed.

2. Experiments with Leydig cells have to be done on 34℃, the temperature is critical for homeostasis of endocrine and exocrine functions of testes.

3. It is necessary to detect the testosterone content in the transfected Leydig cells.

4. P for significance should be capitalized and italicized, please check.

5. The quality of Figure 2B is too poor to be redone.

6. To my knowledge, no significant change was observed by the western blot (bands are saturated).  Change the images or present all western blot images used in densitometric analysis by supplementary figure. This modification will improve the quality of the paper.

7. Although GLOD4 shows a positive effect on Leydig cell maintenance and steroidogenesis, upregulated testosterone synthesis can show negative effects on organs. Do you have any idea or reference for these side effects? If you have, add it to the manuscript.

8. Please discuss the original, and important pioneered results, as well as recent advances in the field focusing on the subject of the study.

Author Response

Reviewer 1

1.Please perform new experiments on primary culture of Leydig cells isolated from animals of different reproductive stages including sexually immature and sexually mature. Then the steroidogenic capacity and activity of Leydig cells at different reproductive stages were observed.

Dear Editor, Our article hoped to explore the effect of GLOD4 on mesenchymal cells, which in turn detected the secretion of testosterone hormone after transfection of the plasmid. The hormone secretion of 12-month-old Leydig cells was not examined routinely.

2.Experiments with Leydig cells have to be done on 34℃, the temperature is critical for homeostasis of endocrine and exocrine functions of testes.

Dear Editor, thank you for pointing this out. We modified the in vitro culture temperature of  Leydig cells.

3.It is necessary to detect the testosterone content in the transfected Leydig cells.

Dear Editor, we examined the secretion of testosterone hormone in transfected Leydig cells in the fifth part of our results

4.P for significance should be capitalized and italicized, please check.

Dear Editor, thank you for pointing this out. We have revised all of that question in the article.

5.The quality of Figure 2B is too poor to be redone.

Dear Editor, thank you for pointing this out. We have reclassified Figure in the hope of making Figure a clearer representation of our results.

6.To my knowledge, no significant change was observed by the western blot (bands are saturated).  Change the images or present all western blot images used in densitometric analysis by supplementary figure. This modification will improve the quality of the paper.

Dear Editor,  we have changed the display density of the image to improve the quality of that image. The zip file(Western-blot images) of our previously submitted manuscript has all the initial images about the western blot. In addition, we have resubmitted the original PDF (Western blot PDF) image of the western blot for your review.

7.Although GLOD4 shows a positive effect on Leydig cell maintenance and steroidogenesis, upregulated testosterone synthesis can show negative effects on organs. Do you have any idea or reference for these side effects? If you have, add it to the manuscript.

Dear Editor, thank you for pointing this out.For this point you mentioned, we have not found any relevant literature reports for the time being, and we also hope that our article can provide a certain theoretical basis for the subsequent relevant studies.

  1. Please discuss the original, and important pioneered results, as well as recent advances in the field focusing on the subject of the study.

Dear Editor, So far, there are not many reports on this gene, and we have described the latest research on this gene in both the Introduction and Discussion sections. And there is only one recent report of this gene in ruminants, which we also discuss.

Reviewer 2 Report

Comments and Suggestions for Authors

Animals-3086574: Expression of GLOD4 gene in the testis of Qianbei Ma goats and its effect on testicular leydig cells. Jinqian Wang et al

General comments:

The manuscript exhibits structural limitations. in particular the sections Mterials and Methods and Discussion need to be clarified.

I think the number of animals (5/group) is too limited. This is a limit to the real impact of results. How authors chose and calculated the sample size?

Please check the text for spelling (e.g. Leydig vs leydig; uppercase and lowercase, word spacing, ect)

Specific comments:

2.2. Cell culture and transfection:

LL. 144-147: “The dispersed cells were then filtered using a 100 mm cell strainer 144

(Biosharp, Shanghai, China)…”: How can the authors be sure that they have isolated only Leydig cells?

L.157: please, correct pasmid

2.4. Total RNA was extracted and reverse-transcribed:

L.169: Total RNA was extracted: please indicate the amount of RNA.

Table 2. Real-time fluorescent primers information: GLOD4 the size of fragment (554 bp) is correct? The optimal amplicon length should be less than 150 bp.

2.9. Cell proliferation analysis: please, add a briefly description of CCK-8 assay

2.10 Steroid analysis: 

LL.235-239: Please indicated the intra- and interassay coefficients.

2.12. Statistics:

LL. 247-251: the models (one way or two way ANOVA) must be provided and described to evaluate the impact of data

Results:

Figure 2: please the entire gel must be shown, not just the two bands (glod4 and actin). The data cannot be evaluated

Discussion: this section could be more efficiently written on the basis of results.

Author Response

Reviewer 2

General comments:

The manuscript exhibits structural limitations. in particular the sections Mterials and Methods and Discussion need to be clarified.

I think the number of animals (5/group) is too limited. This is a limit to the real impact of results. How authors chose and calculated the sample size?

Please check the text for spelling (e.g. Leydig vs leydig; uppercase and lowercase, word spacing, ect)

 Dear Editor, thank you for pointing this out.We have reformatted the format of our article on Leydig cells. In addition, animal numbers of twenty (5 animals/group) are possible for testicular tissue studies in goats, and the following references also have relatively small numbers of ruminants.

Li X, Yao X, Xie H, Deng M, Gao X, Deng K, Bao Y, Wang Q, Wang F. Effects of SPATA6 on proliferation, apoptosis and steroidogenesis of Hu sheep Leydig cells in vitro. Theriogenology. 2021 May;166:9-20. doi: 10.1016/j.theriogenology.2021.02.011. Epub 2021 Feb 20. PMID: 33667862.

Samir H, Nagaoka K, Watanabe G. Effect of kisspeptin antagonist on goat in vitro Leydig cell steroidogenesis. Theriogenology. 2018 Nov;121:134-140. doi: 10.1016/j.theriogenology.2018.07.038. Epub 2018 Aug 14. PMID: 30149259.

Specific comments:

2.2. Cell culture and transfection:

  1. 144-147: “The dispersed cells were then filtered using a 100 mm cell strainer 144

(Biosharp, Shanghai, China)…”: How can the authors be sure that they have isolated only Leydig cells?

 Dear Editor, we cite reference #24 explaining the method of isolating Leydig cells.

L.157: please, correct pasmid

 Dear Editor, thank you for pointing this out.We have identified the word "plasmid", which has the same spelling and conveys the same meaning in the following references.

Li X, Yao X, Xie H, Deng M, Gao X, Deng K, Bao Y, Wang Q, Wang F. Effects of SPATA6 on proliferation, apoptosis and steroidogenesis of Hu sheep Leydig cells in vitro. Theriogenology. 2021 May;166:9-20. doi: 10.1016/j.theriogenology.2021.02.011. Epub 2021 Feb 20. PMID: 33667862.

Ma K, Chen N, Wang H, Li Q, Shi H, Su M, Zhang Y, Ma Y, Li T. The regulatory role of BMP4 in testicular Sertoli cells of Tibetan sheep. J Anim Sci. 2023 Jan 3;101:skac393. doi: 10.1093/jas/skac393. PMID: 36440761; PMCID: PMC9838805.

2.4. Total RNA was extracted and reverse-transcribed:

L.169: Total RNA was extracted: please indicate the amount of RNA.

 Dear Editor, thank you for pointing this out.In the newly submitted manuscript we have added to it, at line 174.

Table 2. Real-time fluorescent primers information: GLOD4 the size of fragment (554 bp) is correct? The optimal amplicon length should be less than 150 bp.

 Dear Editor, thank you for pointing this out.We resubmitted the fragment sizes and primer sequences for GLOD4 in Table 2.

2.9. Cell proliferation analysis: please, add a briefly description of CCK-8 assay

 Dear Editor, thank you for pointing this out.In the newly submitted manuscript we have completed the content in line 237

2.10 Steroid analysis: 

LL.235-239: Please indicated the intra- and interassay coefficients.

Dear Editor, thank you for pointing this out. In the newly submitted manuscript, we added that content at line 247.

2.12. Statistics:

  1. 247-251: the models (one way or two way ANOVA) must be provided and described to evaluate the impact of data

 Dear Editor, thank you for pointing this out. In the new manuscript, we have changed that in line 257

Results:

Figure 2: please the entire gel must be shown, not just the two bands (glod4 and actin). The data cannot be evaluated

 Dear Editor,  we have changed the display density of the image to improve the quality of that image. The zip file(Western-blot images) of our previously submitted manuscript has all the initial images about the western blot. In addition, we have resubmitted the original PDF (Western blot PDF) image of the western blot for your review.

Discussion: this section could be more efficiently written on the basis of results.

Dear Editor, thank you for pointing this out. We have improved the quality of our discussion writing in new submissions.